# Nuclear pore assembly proceeds by an inside-out extrusion of the nuclear envelope

Shotaro Otsuka[1], Khanh Huy Bui[2†], Martin Schorb[2,3], M Julius Hossain[1], Antonio Z Politi[1], Birgit Koch[1], Mikhail Eltsov[1], Martin Beck[2], Jan Ellenberg[1*]

[1]Cell Biology and Biophysics Unit, European Molecular Biology Laboratory, Heidelberg, Germany; [2]Structural and Computational Biology Unit, European Molecular Biology Laboratory, Heidelberg, Germany; [3]Electron Microscopy Core Facility, European Molecular Biology Laboratory, Heidelberg, Germany

**Abstract** The nuclear pore complex (NPC) mediates nucleocytoplasmic transport through the nuclear envelope. How the NPC assembles into this double membrane boundary has remained enigmatic. Here, we captured temporally staged assembly intermediates by correlating live cell imaging with high-resolution electron tomography and super-resolution microscopy. Intermediates were dome-shaped evaginations of the inner nuclear membrane (INM), that grew in diameter and depth until they fused with the flat outer nuclear membrane. Live and super-resolved fluorescence microscopy revealed the molecular maturation of the intermediates, which initially contained the nuclear and cytoplasmic ring component Nup107, and only later the cytoplasmic filament component Nup358. EM particle averaging showed that the evagination base was surrounded by an 8-fold rotationally symmetric ring structure from the beginning and that a growing mushroom-shaped density was continuously associated with the deforming membrane. Quantitative structural analysis revealed that interphase NPC assembly proceeds by an asymmetric inside-out extrusion of the INM.

**\*For correspondence:** jan.ellenberg@embl.de

**Present address:** [†]Department of Anatomy and Cell Biology, McGill University, Montreal, Canada

**Competing interests:** The authors declare that no competing interests exist.

## Introduction

The nuclear pore complex (NPC) is the largest non-polymeric protein complex in eukaryotic cells, embedded in a double membrane called the nuclear envelope (NE), and mediates all macromolecular transport across the NE. The NPC has an octameric structure and is composed of multiple copies of over 30 different proteins termed nucleoporins (Nups) (*Grossman et al., 2012*; *Schwartz, 2013*; *Strambio-De-Castillia et al., 2010*). In metazoan cells NPCs are assembled in two cell-cycle stages, during nuclear assembly post anaphase and during nuclear growth in interphase. Both assembly pathways have distinct properties and are usually referred to as postmitotic and interphase NPC assembly (*Schooley et al., 2012*; *Wandke and Kutay, 2013*). In postmitotic assembly, the double nuclear membrane and the NPC channel assemble concomitantly onto chromatin, and postmitotic formation of import competent nuclei with sealed nuclear membranes and functional NPCs is completed very rapidly within 15 min after anaphase onset (*Dultz et al., 2008*; *Haraguchi et al., 2000*; *Lu et al., 2011*; *Otsuka et al., 2014*).

By contrast, interphase NPC assembly occurs only after the NE is fully sealed in late anaphase. This second assembly mechanism proceeds throughout telo- and interphase resulting in a doubling of the number of NPCs for the next division. In the context of the closed nucleus, NPCs must be formed by an insertion into the NE that fuses the outer and inner nuclear membranes (ONM and INM). Interphase assembly is much slower compared to postmitotic assembly and shows differences

**eLife digest** The nucleus is the compartment within our cells that contains most of our genetic material. It is separated from the rest of the cell by a boundary called the nuclear envelope, which consists of two layers of membrane. All transport in and out of the nucleus has to pass through channels in the envelope, formed by large protein assemblies called the nuclear pore complexes. Each nuclear pore complex is composed of multiple copies of over 30 different proteins termed nucleoporins and there are several hundred proteins per pore.

Before a cell divides in two, the nucleus has to grow and new nuclear pore complexes must be assembled into the double membrane barrier of the nuclear envelope. The assembly process would require the two nuclear membranes to fuse. However, exactly how nuclear pore complexes are assembled has been controversially debated for over 15 years, because no one has directly observed any of the intermediate stages during the assembly process.

Now, Otsuka et al. have captured images of the different steps involved in assembling a nuclear pore complex in a human cell. This was achieved by observing living human cells in which the nucleus was growing and then studying them using advanced techniques such as high-resolution three-dimensional electron tomography and super-resolution microscopy. Otsuka et al. saw dome-shaped bumps or protrusions in the inner nuclear membrane that grew wider and deeper until they fused with the flat outer nuclear membrane. A ring of proteins surrounded the base of these protrusions from the beginning, and the membrane was deformed by a mushroom-shaped collection of proteins. Analysis of the molecules involved in these stages showed that assembly intermediates initially contained nucleoporins that face into the nucleus, and only later were nucleoporins that face into the rest of the cell added to the complex.

The discovery that nuclear pore complexes assemble via an inside-out mechanism in human cells provides a new conceptual framework to interpret existing genetic and biochemical data. The findings also provide a new approach to explore the assembly process in much more detail and ask how nuclear pores first evolved.

in the molecular requirements and the order of recruited components (*D'Angelo et al., 2006*; *Dultz and Ellenberg, 2010*; *Schellhaus et al., 2015*). Several molecular requirements for interphase assembly have been reported in different model systems. Studies using in vitro assembled nuclei with *Xenopus* egg extracts have shown the requirement of RanGTP on both sides of the NE (*D'Angelo et al., 2006*) and the import of Nup153 to recruit the Nup107-160 complex to the INM (*Vollmer et al., 2015*). In mammalian cells, the membrane curvature-sensing domain of Nup133 (*Doucet et al., 2010*), the INM protein Sun1 (*Talamas and Hetzer, 2011*), and the targeting of the transmembrane nucleoporin Pom121 to the INM (*Funakoshi et al., 2011*) have been reported to be required. Although some of these studies as well as a study on the evolution of eukaryotic cells (*Baum and Baum, 2014*) have suggested that interphase NPC assembly may initiate from the nuclear side, how and by what membrane deformation and fusion process NPC assembly takes place has remained enigmatic (*Doucet and Hetzer, 2010*; *Rothballer and Kutay, 2013*). Interestingly, INM deformations have been observed in yeast mutants lacking several nucleoporins, membrane proteins Apq12 and Brr6, and the AAA-ATPase VPS4 and, while sometimes interpreted as pleiotropic consequences of transport defects, have also been suggested to be involved in nucleoporin quality control or NPC assembly (*Chadrin et al., 2010*; *Hodge et al., 2010*; *Makio et al., 2009*; *Meszaros et al., 2015*; *Murphy et al., 1996*; *Scarcelli et al., 2007*; *Webster et al., 2014*; *Wente and Blobel, 1993*). However, it has remained unclear how NPC assembly takes place in wild-type cells and what the normal assembly intermediates might look like. Pioneering studies that used in vitro assembled and inhibitor treated nuclei (*Goldberg et al., 1997*) could unfortunately not establish the physiological nature of the partial NPC structures since they only examined the cytoplasmic side of the NE and were not able to analyze INM deformations.

Despite this significant amount of indirect evidence and several competing hypotheses for interpreting it regarding NPC assembly (*Rothballer and Kutay, 2013*), progress in the field has been slow largely due to the experimental challenge of capturing the rare and sporadic interphase NPC

assembly events and imaging them at single pore resolution in order to reliably distinguish newly-assembling from already-formed NPCs (*D'Angelo et al., 2006*; *Dultz and Ellenberg, 2010*). To overcome this challenge and study the mechanism of interphase assembly in whole cells more effectively, we focused on the NPC-poor NE islands present in telophase nuclei that are populated with NPCs during nuclear expansion in the G1 phase of the cell-cycle (*Maeshima et al., 2006*). These islands result from the so called 'core regions' where nuclear membrane sealing is locally delayed in mitosis due to removal of dense spindle microtubules from the DNA surface (*Vietri et al., 2015*) and therefore largely devoid of postmitotic NPC assembly, resulting in a low NPC density in the membrane of the core regions (*Dechat et al., 2004*; *Haraguchi et al., 2000*). Core regions therefore provide an almost 'virgin' double membrane surface, where *de novo* interphase NPC assembly is easier to observe. By systematically recording electron tomograms of core regions at different times of nuclear growth, using correlation with live imaging to determine the precise cell-cycle stage of each cell, we were indeed able to reliably capture intermediates of interphase NPC assembly. Three-dimensional (3D) analysis of temporally ordered intermediates revealed that interphase NPC assembly proceeds by an inside-out INM evagination followed by fusion with the flat ONM. Averaging the structure of assembly intermediates at the same stage of membrane deformation showed that an eightfold symmetric nuclear ring underneath the INM already surrounds the base of the earliest detectable evaginations and that a mushroom-shaped density appears to drive the membrane deformation until fusion with the ONM.

## Results and discussion

### Correlative electron tomography captures intermediates of interphase NPC assembly

Deformation and fusion of the nuclear membranes that must be present during interphase NPC assembly can only be reliably detected by high-resolution 3D electron microscopy (EM). To target such EM observations, we established an assay that allowed us to estimate the position of the core region in the NE of telophase and G1 nuclei at any time during nuclear expansion post anaphase. To this end, we used 3D live confocal time-lapse imaging of the core marker Lap-2α tagged with YFP (*Dechat et al., 2004*) together with the chromatin marker histone 2B tagged with mCherry (*Figure 1—figure supplement 1A*). 3D reconstruction of the core region surface in late anaphase allowed us to calculate the core regions at later times in the cell-cycle by modeling it onto the overall growth of the nuclear surface measured using histone 2B (*Figure 1—figure supplement 1B–H*). With this assay in hand, we then systematically imaged live cells after exiting mitosis on EM compatible sapphire disks with carbon coated landmarks (*Figure 1A*), and natively fixed them by rapid high pressure freezing at defined times during G1 nuclear expansion. After cryo-substitution, we acquired high-resolution electron tomograms from sections cut through the core regions. The single cell correlation with live imaging allowed us to precisely determine the stage of nuclear expansion of each cell sampled by electron tomography and therefore temporally register all our samples (*Figure 1A*, *Figure 1—figure supplement 2*).

In the resulting 158 tomograms, we consistently found approximately 50 nm evaginations of the INM (*Figure 1B*, *Table 1*, and *Video 1*) filled with electron dense material, that were clearly distinct from the ~200 nm nuclear egress structures transporting ribonucleoproteins and viruses reported recently (*Mettenleiter et al., 2013*; *Speese et al., 2012*). Immuno-EM showed that the evaginations were specifically enriched with at least one of the nucleoporins recognized by mAb414 (Nup62, Nup153, Nup214, and Nup358) (*Figure 1C,D*, and *Figure 1—figure supplement 3A*), suggesting that they are pore assembly intermediates. Similar INM evaginations filled with electron dense material were also found in cryo-electron tomograms of vitrified isolated NEs (*Figure 1E*), ruling out that they are artifacts of dehydration, heavy metal staining or resin-embedding during cryo-substitution, and demonstrating that they are stable membrane structures that persist even after in vitro isolation of the NE (*Bui et al., 2013*; *Ori et al., 2013*). Indistinguishable evaginations of the INM were also observed in U2OS (human bone osteosarcoma epithelial) and NRK (normal rat kidney) cells (*Figure 1—figure supplement 3B*), ruling out that their occurrence is cell type, cancer or species specific.

 

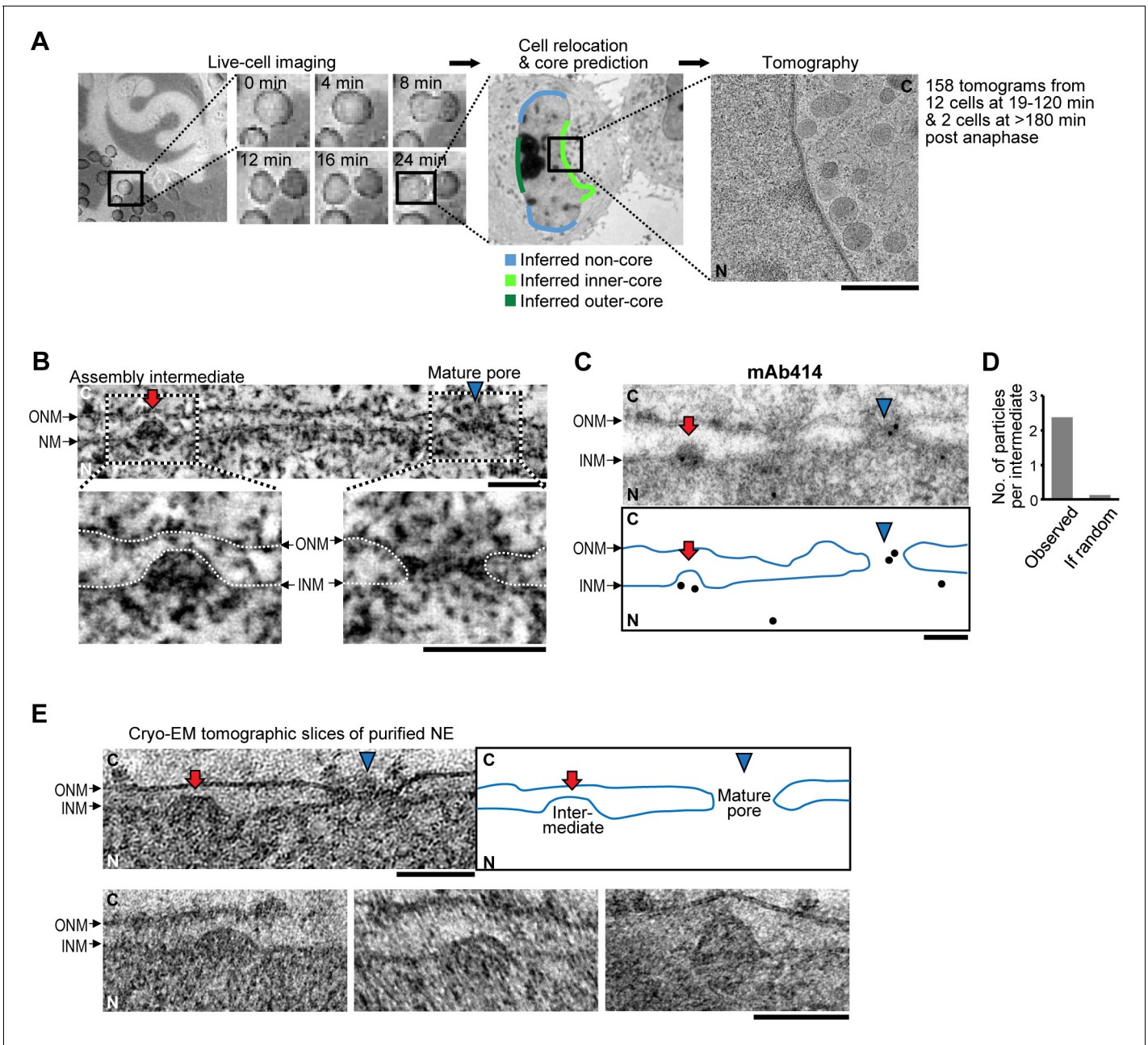

**Figure 1.** Interphase assembly intermediates of nuclear pore complexes (NPCs). (A) Correlative live-cell imaging with electron microscopy (EM). Cell-cycle progression of HeLa cells was monitored by confocal microscopy and the same cell was subjected to electron tomography. Tomograms were collected from different regions of the nuclear envelope (NE). Inferred non-core, inner-core and outer-core regions are indicated in light blue, light green and dark green, respectively. C, cytoplasm; N, nucleoplasm. Scale bar, 1 µm. (B) An electron tomographic slice of the NE. An assembly intermediate and a mature pore are indicated by a red arrow and a blue arrowhead, respectively. Insets show enlarged images in which membranes are traced by white dotted lines. ONM, outer nuclear membrane; INM, inner nuclear membrane. Scale bars, 100 nm. (C,D) Immuno-EM with mAb414 antibody and 10 nm-gold particles. (C) The profile of the NE and the positions of gold particles are denoted in the bottom panel. A mature pore and an intermediate are indicated as in (B). Scale bar, 100 nm. (D) The number of gold particles per assembly intermediate ('observed') and the one calculated assuming a random distribution of the particles ('if random'). 31 particles were found on 13 intermediates, whereas the random distribution estimated 1.6 particles to be on 13 intermediates. The p-value (probability that the distribution is due to chance alone) <$10^{-100}$; a chi-square goodness of fit test. (E) Cryo-EM tomographic slices of isolated NEs of HeLa cells. A mature pore and an intermediate are indicated as in (B). Other examples of intermediates are also indicated. Scale bars, 100 nm.

The following figure supplements are available for figure 1:

**Figure supplement 1.** Estimation of core regions.

*Figure 1 continued on next page*

*Figure 1 continued*

**Figure supplement 2.** Live-cell and EM images of cells analyzed by EM tomography.

**Figure supplement 3.** Galleries of interphase NPC assembly intermediates.

## Assembly intermediates grow inside-out

To test if the evaginations displayed a progression of structural changes consistent with the formation of mature NPCs, we analyzed their membrane shape in cells captured at different time points after the completion of postmitotic nuclear assembly. Quantitative analysis of 135 INM evagination membrane profiles from a time course of cells captured at 19, 28, and 53 min post anaphase revealed that evaginations progressively grow inside-out (*Figure 2*). Evagination depth increased significantly from 16 to 22 nm within 9 min (19 to 28 min post anaphase; *Figure 2C*) and evagination diameter continuously and significantly increased from 51 to 58 nm within 34 min (19 to 53 min post anaphase; *Figure 2D*). Among the 279 total evaginations we found in 154 $\mu m^2$ NE surface area (*Table 1*), we capture only five ONM/INM fusion events (*Figure 2*), where the INM evagination had reached the flat ONM surface. These fusion intermediates had an average evagination depth of 28 nm, similar to the ONM/INM distance and an average diameter of 61 nm, intermediate between late evaginations and mature nuclear pores (*Figure 2C,D*), as expected for NPC assembly. The low number of fusion intermediates indicates that the fusion step must be very short-lived.

## Abundance of intermediates matches increase in mature pores during nuclear growth

If the assembly intermediates we observed mature into fully assembled NPCs, their abundance should quantitatively explain the increased number of mature pores observed after nuclear expansion. To address this, we quantified the changes in density of intermediates and mature pores over time in EM tomograms of a time course of 12 cells correlatively fixed from 19 to 120 min post anaphase (*Figure 3* and *Figure 1—figure supplement 2*, and *Table 1*). This data showed that assembly intermediates are most abundant in core regions during the first hour, when this sealed membrane area still has a low density of mature pores due to the lack of postmitotic assembly (*Figure 3B*). By

**Table 1.** Summary of EM tomography. A data table shows the surface area of the NE analyzed by EM tomography and the number of mature pores, assembly intermediates, and the outer and inner nuclear membrane (ONM and INM) fusion events found in each cell at a different time point after anaphase onset. The data obtained in non-core, inner- and outer-core regions are indicated separately. In total, 154 $\mu m^2$ NE surface area was analyzed, and 279 intermediates and 1322 mature pores were found.

| | Time after anaphase onset (min) | 19.2 | 24.4 | 28.4 | 36.3 | 42.0 | 53.2 | 61.0 | 65.6 | 73.6 | 82.9 | 100 | 116 | >180 | >180 |
|---|---|---|---|---|---|---|---|---|---|---|---|---|---|---|---|
| Non-core | Analyzed surface area ($\mu m^2$) | 5.29 | 4.04 | 5.06 | 3.99 | 5.17 | 5.54 | 4.04 | 3.72 | 3.06 | 3.73 | 3.32 | 3.17 | 4.03 | 2.80 |
| | Number of mature pores | 82 | 45 | 48 | 42 | 62 | 51 | 53 | 33 | 33 | 39 | 33 | 31 | 45 | 31 |
| | Number of intermediates | 2 | 4 | 6 | 3 | 5 | 8 | 3 | 3 | 2 | 1 | 5 | 5 | 4 | 1 |
| | Number of ONM/INM fusion | | | | | | | | | | | | | | |
| Inner-core | Analyzed surface area ($\mu m^2$) | 5.77 | 5.06 | 4.16 | 4.74 | 6.75 | 4.33 | 3.61 | 4.57 | 4.07 | 3.53 | 4.61 | 3.46 | | |
| | Number of mature pores | 24 | 16 | 3 | 18 | 30 | 16 | 34 | 27 | 27 | 39 | 53 | 32 | | |
| | Number of intermediates | 21 | 13 | 12 | 8 | 35 | 12 | 1 | 3 | 4 | 3 | 2 | 3 | | |
| | Number of ONM/INM fusion | | | | | 2 | | | | 1 | | | | | |
| Outer-core | Analyzed surface area ($\mu m^2$) | 4.22 | 4.15 | 3.83 | 2.55 | 4.78 | 4.41 | 3.59 | 3.68 | 2.71 | 2.67 | 2.77 | 2.68 | | |
| | Number of mature pores | 29 | 40 | 27 | 17 | 44 | 35 | 38 | 28 | 30 | 34 | 25 | 28 | | |
| | Number of intermediates | 23 | 10 | 19 | 9 | 13 | 11 | 1 | 4 | 1 | 3 | 5 | 1 | | |
| | Number of ONM/INM fusion | | | | | 2 | | | | | | | | | |

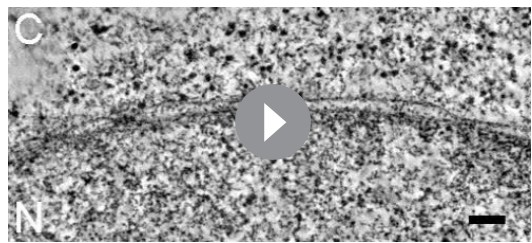

**Video 1.** EM tomographic slices of the nuclear envelope of a cell at 53 min post anaphase. One of the mature pores and an assembly intermediate are indicated by blue and red arrows, respectively. Scale bar, 100 nm.

contrast and as expected, non-core regions already exhibited a high density of mature pores that arose from postmitotic NPC assembly (*Figure 3B*). Later than one hour post anaphase, the ratio of assembly intermediates to mature pores in the core regions had equilibrated to a similar level as found in non-core regions or fully grown nuclei (*Figure 3A,B*), indicating that a period of frequent interphase pore assembly events during the first hour of G1 nuclear expansion populates the core regions of the NE with NPCs until the steady state interphase density is reached.

To test if the high abundance of assembly intermediates in core regions quantitatively explains the number of mature pores found in the same region at later times, we formulated a simple mathematical model for nuclear pore assembly. In this model, assembly intermediates are produced, enter a maturation phase, become mature pores after a typical maturation time, and are ultimately degraded (*Figure 3—figure supplement 1A* and Materials and methods). NPC production and degradation rates were estimated from the measured steady state density of 11 NPCs/$\mu m^2$ (*Figure 3B*) and a reported NPC lifetime of ~40 hr in cells with a similar cell-cycle duration (*Rabut et al., 2004*; *Schwanhausser et al., 2011*) and are in line with the rare and rapid pore disassembly events that have been observed in mammalian cells (*Dultz and Ellenberg, 2010*). We modeled different scenarios for the appearance of intermediates (*Figure 3—figure supplement 1B,C*). Provided that intermediates start to be initiated shortly after anaphase onset the model fits the experimental data of the core regions from 19–120 min post anaphase well (*Figure 3C,D*, and *Figure 3—figure supplement 1D,E*, Variant 2, 3), confirming that the abundance of assembly intermediates we observed at the beginning of nuclear growth quantitatively explains the number of mature NPCs observed one hour later. For the best model, where the majority of intermediates are initiated 10 starting minutes after anaphase onset (*Figure 3—figure supplement 1B,C*, Variant 3) we can estimate the typical maturation time for interphase assembly to be 44 min (95% confidence interval [41–50]). Similar average maturation time was obtained for alternative models where maturation steps are explicitly included (*Figure 3—figure supplement 1F,G*) or where the maturation time has a broader distribution (*Figure 3—figure supplement 1H,I*), demonstrating the robustness of our results. The obtained maturation time is in good agreement with previous reports based on fluorescence microscopy (~25 min, *D'Angelo et al., 2006*; ~60 min, *Dultz and Ellenberg, 2010*).

## Abundance of inside-out evaginations accounts for NPC formation throughout interphase

It is important to note that inside-out evaginations were present at lower density in non-core regions (*Figure 3B* 'Non-core') and that we found no significant difference in the increase in depth or diameter of evaginations between non-core and core regions during G1 expansion (*Figure 2E,F*). This rules out that inside-out assembly intermediates are specific to the core regions, or that core regions are delayed in their maturation. In addition, identical evaginating structures were also found at low density in fully grown nuclei sampled at later cell-cycle stages (*Figure 3B* 'Mature NE') indicating that the inside-out assembly mechanism is not specific to G1 but occurs throughout interphase.

Assuming the maturation time of 44 min, our model shows that the steady state abundance of assembly intermediates we observed in non-core regions and interphase nuclei would be sufficient to maintain the constant NPC density during nuclear growth in interphase that we observed (*Figure 3B* "model", and *Figure 3—figure supplement 1*, *2*) consistent with previous reports (*Dultz and Ellenberg, 2010*; *Maeshima et al., 2010*). Taken together these results suggest that the assembly intermediates are present across the NE surface, and that the kinetics of NPC assembly are similar across the nuclear surface and throughout interphase. Most importantly, the abundance of intermediates can quantitatively explain both the increase in mature NPCs in the core regions

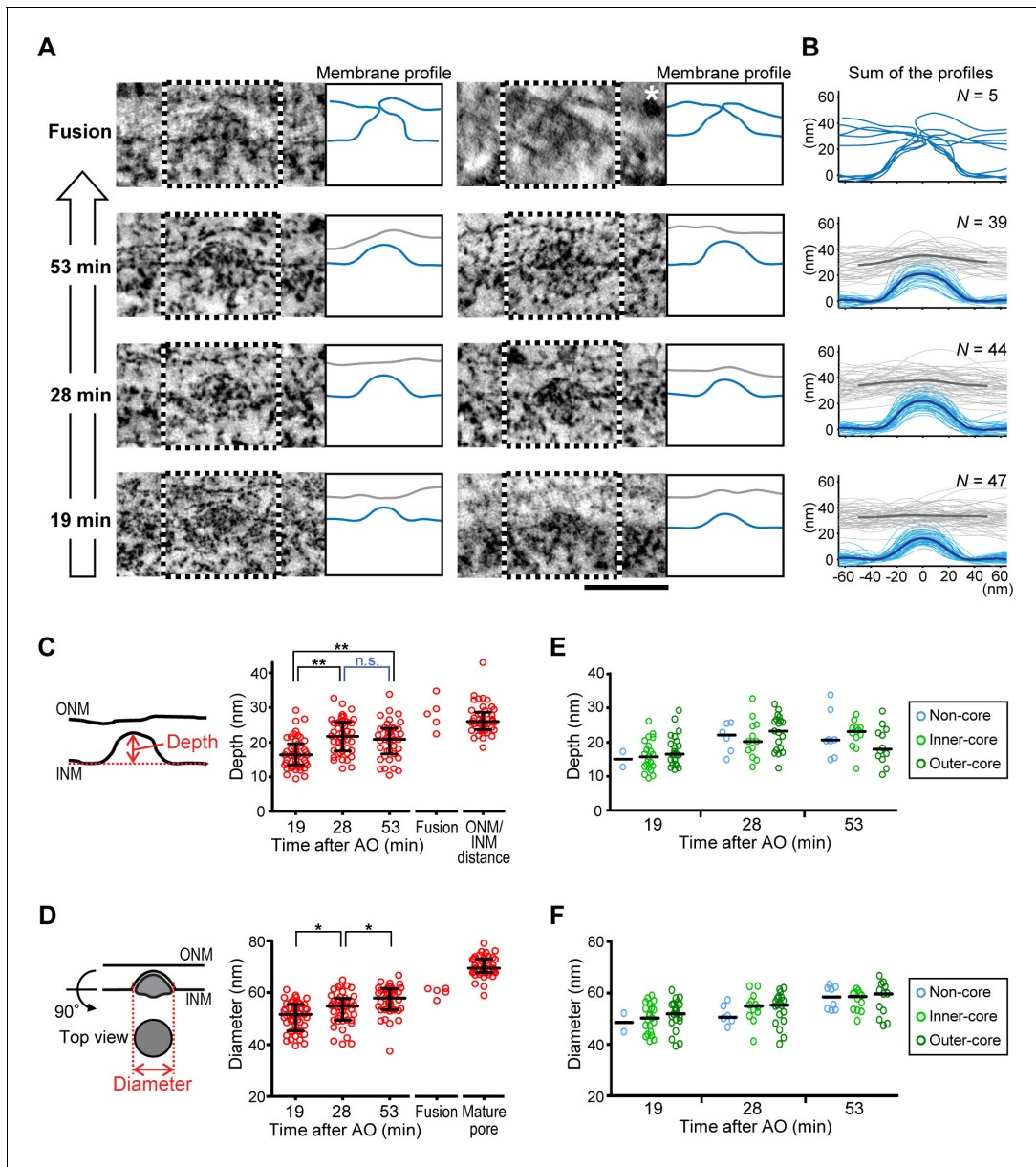

**Figure 2.** Quantitative structural comparison of assembly intermediates. (A) Electron tomographic slices of assembly intermediates in cells captured at 53, 28, and 19 min after anaphase onset (AO) and ONM/INM fusion events. Profiles of ONM (gray) and INM (blue) in black and white boxes on EM images are depicted in the right panels. For the fusion, ONM is also depicted in blue. The image marked with a white asterisk was acquired on a differently embedded sample for enhancing membrane contrast (see Materials and methods). Scale bar, 100 nm. (B) Membrane profiles of all the fusion events and intermediates at selected time points (53, 28, and 19 min). The bold lines indicate the averaged profiles. (C–F) Quantification of the evagination depth of INM (C,E) and the diameter of intermediates (D,F) as indicated by red bidirectional arrows in the left panels. (C,D) The plots are from 47, 44, and 39 intermediates at 19, 28, and 53 min, respectively, 5 ONM/INM fusions, and 45 mature pores. The ONM/INM distance was quantified near mature pores (C). The median is depicted as a horizontal line and the whiskers show the 25th and 75th percentiles. *p<0.02, **p<0.001; unpaired t-tests. (E,F) The depth and the diameter of intermediates in non-core, inner-core, and outer-core regions were indicated in light blue, light green and dark green, respectively. The median is depicted as a horizontal line. No statistical difference of the intermediate shape was observed between different regions of the NE at each time point (p>0.1; unpaired t-tests).

The following source data is available for figure 2:

**Source data 1.** Depth and diameter values used for *Figure 2C–F*.

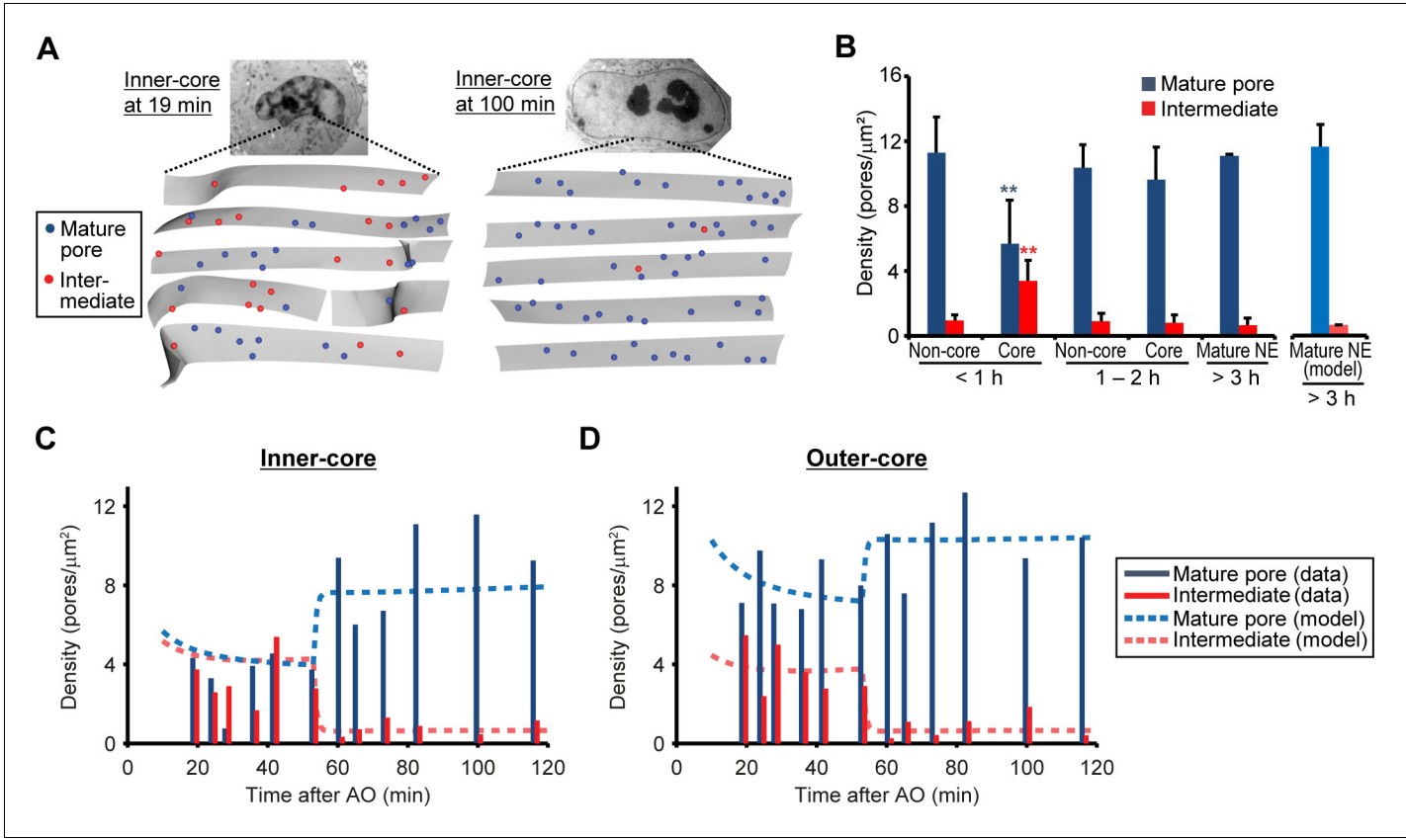

**Figure 3.** Abundance of mature pores and assembly intermediates at different cell-cycle stages. (A) Measurement of nuclear pore density. Gray sheets are the NEs segmented from EM tomograms. Blue and red dots indicate the positions of mature pores and intermediates, respectively. Inner-core regions of cells at 19 and 100 min post anaphase are shown as examples. (B) Density of mature pores (dark blue) and intermediates (red) in non-core and core (inner- plus outer-core) regions of cells at different times. Error bars represent the s.d. from 6, 6, and 2 cells at <1, 1–2, and >3 hr post anaphase, respectively. **p<0.001; unpaired t-tests of the density difference of mature pores (blue) and intermediates (red) between core regions at <1 hr and the others. The modeled density of mature pores (light blue) and intermediates (light red) at >3 hr is also indicated (see *Figure 3—figure supplement 1* and Materials and methods for details). (C,D) Density of mature pores and intermediates in inner- (C) and outer-core (D) regions of cells at different time points. 3–7 tomograms were obtained in each region at each time point (data are summarized in *Table 1*). Dashed lines indicate the modeled density of mature pores and intermediates.

The following source data and figure supplements are available for figure 3:

**Source data 1.** Density values used for *Figure 3B–D*.
**Figure supplement 2—Source data 1.** Surface area values used for *Figure 3— figure supplement 2C*.
**Figure supplement 1.** Modeling the density of nuclear pores.
**Figure supplement 2.** Nuclear surface area measurement for the modeling.

during the rapid nuclear expansion in G1 as well as the homeostatic NPC assembly during nuclear growth later during interphase.

## Live imaging of the core region reveals progressive maturation of intermediates

The high abundance of intermediates in the core region during the first hour of G1 (*Figure 3C,D*) and their progressive increase in depth and diameter during this time (*Figure 2*) indicates that the assembly process is relatively synchronous while the NPC-poor core region is populated to the same

density as the rest of the nuclear surface (*Figure 3B*). We should therefore be able to observe the maturation of assembly intermediates directly, by imaging nucleoporin accumulation in the NPC-poor core region in real time. To test this, we performed fast 3D live confocal time-lapse imaging and monitored the concentration of GFP-tagged nucleoporins in the core region during the first hour of G1, the same time window we observed by correlative EM (*Figure 4A*). Since the intermediates in inner- and outer-core regions grow in a similar manner (*Figure 2E,F*) and show similar abundance (*Figure 3C,D*), we measured only the inner-core region where the proportion of intermediates to mature pores is much higher than in the outer-core region.

Since our EM analysis suggested an inside-out mechanism, we selected the nuclear and cytoplasmic ring component Nup107 (*Belgareh et al., 2001*) and the cytoplasmic filament component Nup358 (*Wu et al., 1995*; *Yokoyama et al., 1995*) as candidate nucleoporins for genomic GFP-tagging (*Figure 4—figure supplement 1*) and live imaging. As predicted, the accumulation kinetics differed substantially between core and non-core regions of the NE for both proteins (*Figure 4B,C*). Since the non-core regions and inner-core regions contained 8% and 50% assembly intermediates in early G1 respectively (*Figure 3B,C*), we used them to determine the postmitotic and interphase rates of accumulation that explain the different accumulation kinetics of the core region for both Nups resulting from the combined rates (*Figure 4B,C*, and Materials and methods). Kinetic comparison of interphase accumulation of both Nups in the core region clearly revealed that Nup107, a component of the nuclear and cytoplasmic ring, is recruited very early ($t_{1/2}$ = 15 min, see Materials and methods for details), while Nup358, a component of the cytoplasmic filaments, is recruited only after a significant lag phase ($t_{1/2}$ = 51 min) (*Figure 4D*). The kinetically distinct and continuously increasing accumulation of two components of the NPC in the core region strongly support a maturation process of assembly intermediates into full pores. In addition, the late recruitment of the cytoplasmic Nup358 is consistent with an inside-out assembly mechanism on the molecular level.

## Single pore assembly intermediates contain Nup107 but not Nup358

The kinetic analysis of bulk Nup accumulation across the inner-core region predicts that single NPC intermediates in early G1 cells should contain Nup107 but not Nup358. To resolve single intermediates with bi-molecular labeling, we used live imaging to stage cells in G1 and then correlatively performed two-color super-resolution imaging using stimulated emission depletion (STED) microscopy with specific antibodies to detect Nup107 and Nup358 (Materials and methods). This analysis indeed revealed many pore-sized discrete localizations of Nup107 in optical sections of the inner-core region NE in early G1 cells, which did not have significant Nup358 labeling (*Figure 5A,B* '24 min, inner-core'), while non-core regions in the same nucleus contained only double labeled localizations with Nup358 appearing on top of the Nup107 labeling on the outside of the NE (*Figure 5A,B* '24 min, non-core'), indicative of mature pores. After G1 expansion, also the inner-core region had almost only double-labeled structures (*Figure 5A,B* '108 min'), consistent with the maturation of intermediates into mature pores. Quantification of the signal in segments along the NE profile allowed us to estimate the frequency of intermediates by the ratio of Nup107/Nup358 (*Figure 5C*), showing that they are specific to the core region and occur only transiently during the first hour of G1. These results are fully consistent with the EM observations that interphase NPC assembly intermediates populate the core region of NEs with an abundance that matches the number of mature pores found in this region an hour later after nuclear expansion (*Figure 3A,B*), and suggest that cytoplasmic nucleoporins such as Nup358 are only recruited at the end of the maturation process, presumably after the growing INM evagination has fused with the ONM.

## INM evaginations are surrounded by an 8-fold rotationally symmetric nuclear ring and filled with a mushroom-shaped cap

Nup107 is a component of the eight-fold rotationally symmetric cytoplasmic and nuclear rings of the NPC. Its early presence in assembly intermediates and their inside-out nature suggested that the nucleoplasmic ring might be one of the first structural elements to form during NPC assembly. To test this, we performed particle averaging of the electron densities of single INM evaginations isolated from tomograms, staged by time during G1 and picked by similarity of their membrane profile depth and diameter. Averaging of 11–36 intermediates revealed a ring structure composed of eight regularly spaced subunits underneath the INM (*Figure 6B* and *Figure 6—figure supplement 1*),

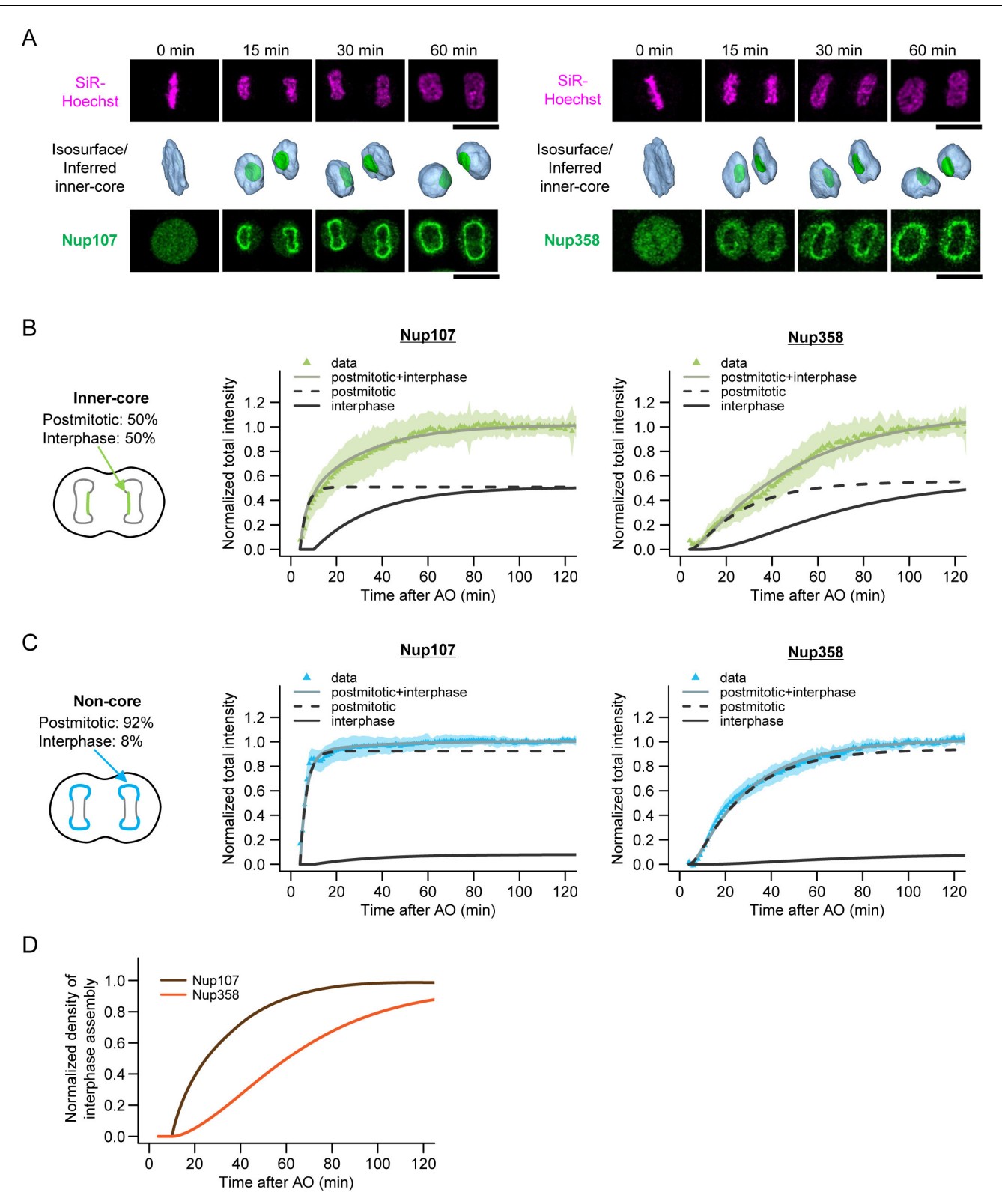

**Figure 4.** Live imaging of nuclear pore assembly in core regions. (**A**) Time-lapse three-dimensional (3D) imaging of GFP-Nup107 and GFP-Nup358 genome-edited cells. DNA was stained with silicon–rhodamine (SiR) Hoechst (*Lukinavicius et al., 2015*). Single confocal sections of SiR and GFP channels are shown in the top and bottom panels, respectively. Segmented chromosomes (light blue) and inferred inner-core regions (green) are shown in the middle panels. Time after anaphase onset is indicated. Scale bars, 20 μm. (**B,C**) Quantification of Nup107 (left) and Nup358 (right) assembly in

*Figure 4 continued on next page*

*Figure 4 continued*

inner-core (**B**) and non-core (**C**) regions. The population of postmitotic and interphase NPC assembly measured in *Figure 3A–C* is indicated in the left panels. Total intensities of Nup107 (left) and Nup358 (right) were quantified, normalized, and fitted with a sequential model of NPC assembly that allows for different rate constants for postmitotic and interphase assembly, respectively (*Equations 16–18* in Materials and methods). Dots and shaded areas represent the average and s.d. of measurements from 30 cells for Nup107 and 25 cells for Nup358, respectively. Black dashed and solid lines indicate the postmitotic and interphase assembly kinetics and gray solid lines show the combined kinetics. (**D**) Normalized densities of interphase Nup107 (brown) and Nup358 (orange) assembly. The density was measured by dividing the intensity obtained in (**B**) by the nuclear surface area.

The following source data and figure supplement are available for figure 4:

**Source data 1.** Intensity values used for *Figure 4B,C*.
**Figure supplement 1.** Characterization of genome-edited cell lines expressing GFP-Nup107 and GFP-Nup358.

which was strikingly similar to the nuclear ring of the mature NPC (compare top views (ii) of the mature pore and intermediates in *Figure 6B*) (*Bui et al., 2013*; *Maimon et al., 2012*). These eight-fold symmetric rings were already present in the shallowest evaginations we could detect during early G1 and could also be seen in individual tomograms (*Figure 6A,B*, and *Figure 6—figure supplement 1*). Interestingly, side views of the averaged particles revealed a progressive growth also of the mushroom-shaped protein density, whose cap closely matched the growing membrane evagination in depth and diameter and whose stalk was located centrally inside the nuclear ring and grew in length as the cap moved towards the outer membrane (*Figure 6B*). The structure revealed by

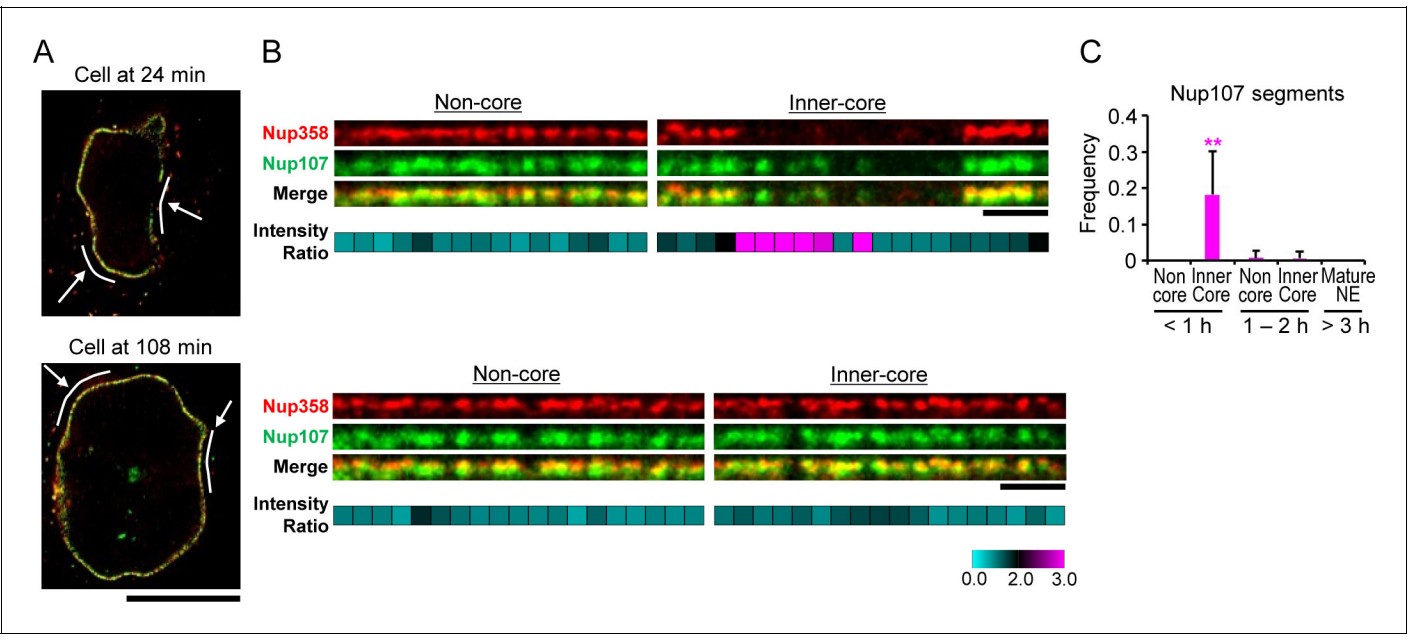

**Figure 5.** Stimulated emission depletion (STED) imaging of assembly intermediates. GFP-Nup107 genome-edited cells were stained with anti-GFP and anti-Nup358 antibodies. (**A**) STED images of cells at 24 and 108 min after anaphase onset. Scale bar, 10 µm. (**B**) Flattened and enlarged images of the inferred non-core and inner-core regions indicated by white lines and arrows in (**A**). The intensity ratios of Nup107 to Nup358 were quantified in every 300 nm segments along the NE and are shown in cyan-black-pink heat maps in the bottom panels. Scale bars, 1 µm. (**C**) The frequency of the segments with the Nup107/Nup358 ratio of >2.0 in non-core and inner-core regions at different times. The data are from 14 cells at <1 hr, 6 cells at 1–2 hr, and 4 cells at >3 hr after anaphase onset. Error bars represent the s.d.. **p<0.001; unpaired t-tests of the frequency difference between the inner-core region at < 1 hr and the others.

The following source data is available for figure 5:

**Source data 1.** Frequency values used for *Figure 5C*.

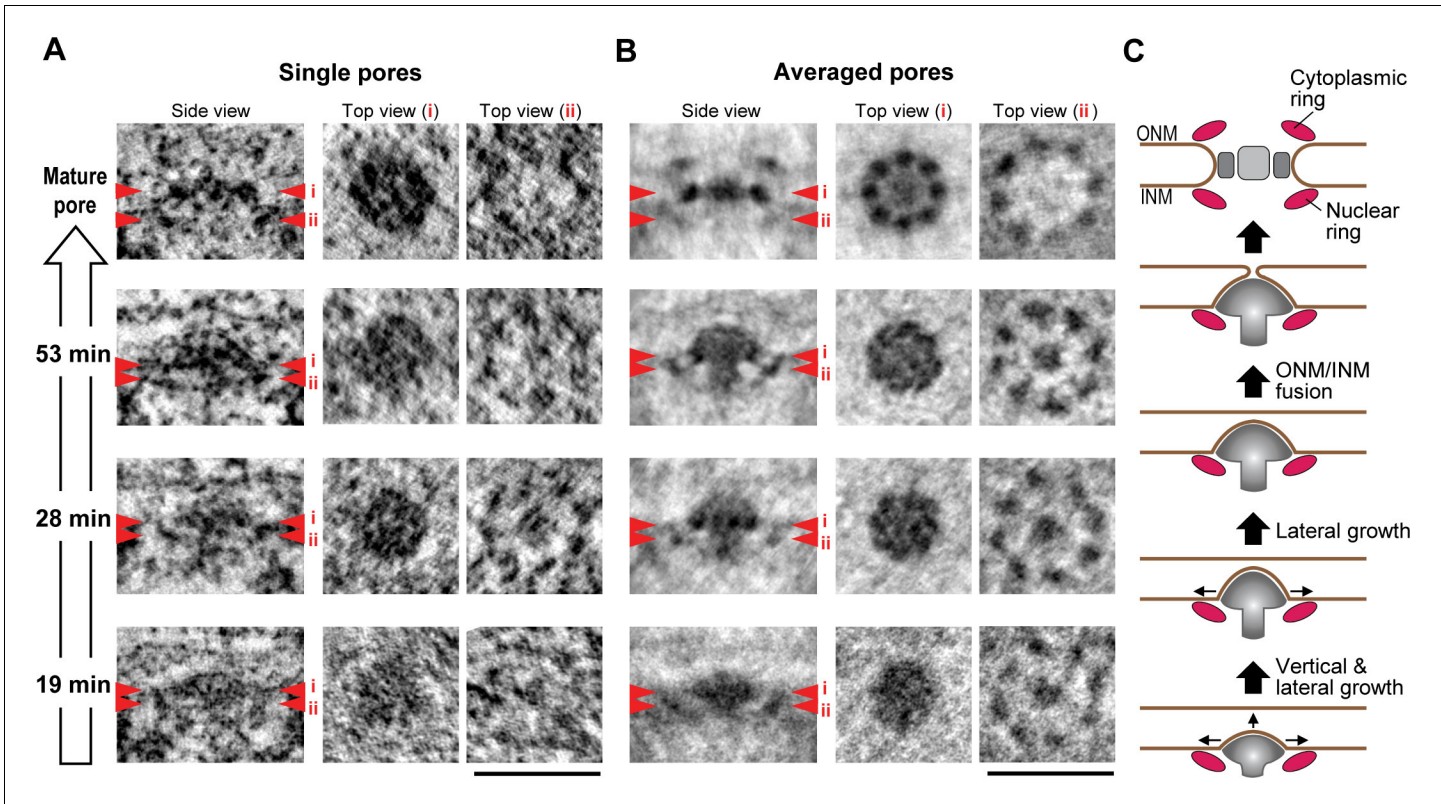

**Figure 6.** 3D structural comparison of assembly intermediates. (**A,B**) Electron tomographic slices of single (**A**) and averaged (**B**) mature pores and intermediates at selected time points (53, 28, and 19 min). The averaged images are from 36 mature pores and 14, 11, and 24 intermediates picked by similarity of their membrane profile depth and diameter at 53, 28, and 19 min, respectively. Red arrowheads i and ii on side-view images indicate the locations of the planes which are inclined at 90° in top views i and ii. The 8-fold symmetric rings observed in top views i and ii of the averaged mature pore are the spoke ring and the nuclear ring complexes, respectively (*Bui et al., 2013*; *Maimon et al., 2012*). Scale bars, 100 nm. (**C**) A schematic model of interphase nuclear pore assembly. The assembly intermediate is comprised of the nuclear ring and the central mushroom-shaped density. The assembly of the mushroom drives the INM deformation and it grows progressively inside-out. Once the ONM and INM fuse, it undergoes rapid and drastic structural rearrangements and finally becomes a mature pore.

The following figure supplement is available for figure 6:

**Figure supplement 1.** Stability of the subtomogram averaging.

particle averaging of assembly intermediates is thus consistent with the observation that Nup107 (a component of the nuclear ring) assembles at an early stage and reveals a very interesting mushroom shaped density that might be the driving force of the membrane evagination.

Taken together, we can conclude that the observed INM evaginations represent partially assembled NPCs, which contain minimally the nuclear ring with Nup107 and at least one of the central or nuclear O-glycosylated FG-repeat nucleoporins labeled by mAb414, Nup62 and/or Nup153, do not contain Nup358, and are unlikely to contain the cytoplasmic filament base protein Nup214 due to their inside out nature and the lack of Nup358.

## Inside-out model of interphase NPC assembly

Our quantitative structural analysis of the membrane profiles and protein densities of a temporally ordered series of NPC assembly intermediates allowed us to reveal a novel mechanism for NPC biogenesis in intact nuclei of interphase cells by an inside-out extrusion of the NE (*Figure 6C*). The first clearly detectable NPC intermediate is a shallow INM evagination surrounded at its base by an 8-fold rotationally symmetric nuclear ring complex, in whose center a dome-shaped density with a short stalk is embedded into the INM evagination. Subsequently, this shallow dome matures into a

curved mushroom cap, always in direct contact with the growing evagination of the INM and supported by an elongating stalk on the nucleoplasmic side located in the center of the nuclear ring. We speculate that the mushroom-shaped density may use the membrane-attached nuclear ring to determine the site of NPC formation. It is further tempting to speculate that the mushroom-shaped density, potentially through connections to the nuclear ring, might generate the mechanical force needed for INM deformation and eventual fusion with the ONM. Interestingly, the mushroom-shaped structure is clearly distinct from the scaffold architecture of the mature NPC, indicating that interphase assembly cannot be explained by a simple collection of NPC subcomplexes over time but likely involves major structural rearrangements. Given that it had so far been unclear how interphase NPC assembly occurs, this inside-out extrusion mechanism, demonstrated in situ in human cells, provides a new framework to interpret existing genetic (yeast) and biochemical (*Xenopus*) data and to investigate the detailed molecular mechanism regulating the assembly process in the future.

## Materials and methods

### Cell culture

Wildtype HeLa kyoto cell line was from Prof. Narumiya in Kyoto University (RRID: CVCL_1922), and the genome was sequenced previously (*Landry et al., 2013*). Wildtype NRK (RRID:CVCL_3758) and U2OS (RRID:CVCL_0042) cell lines were purchased from ATCC (Wesel, Germany). HeLa and NRK cells were grown in Dulbecco's Modified Eagle's Medium (DMEM) (Sigma Aldrich, St. Louis, MO) supplemented with 10% fetal calf serum (FCS), 2 mM glutamine, 1 mM sodium pyruvate, and 100 µg/ml penicillin and streptomycin. U2OS cells were grown in McCoy's 5A Medium (Sigma Aldrich) supplemented with 10% fetal calf serum (FCS), 1X non-essential amino acids solution (Gibco, Waltham, MA), 5 mM glutamine, 1 mM sodium pyruvate, and 100 µg/ml penicillin and streptomycin. A plasmid carrying Lap-2α fused with YFP (*Dechat et al., 2004*) was introduced into HeLa cells with the transfection reagent, Fugene6 (Promega, Madison, WI), according to the manufacturer's protocol. A HeLa cell line stably expressing histone H2b-mCherry (*Neumann et al., 2010*) was maintained at 500 ng/ml puromycin (Invitrogen, Carlsbad, CA). The mycoplasma contamination was tested by PCR every 2 or 3 months and was always negative. Cells were cultured on 2-well Lab-Tek Chambered Coverglass (Thermo Fisher Scientific, Waltham, MA) for live-cell imaging. For correlative light–electron microscopy, cells were grown on sapphire disks (0.05 mm thick, 3 mm diameter; Wohlwend GmbH, Sennwald, Switzerland), which had been carbon-coated in order to relocate cells on electron microscopy (EM) grids, and synchronized by double thymidine arrest (*Harper, 2005*).

### Live-cell imaging

At least 30 min before imaging, the medium was replaced by imaging medium (IM; $CO_2$-independent medium without phenol red (Invitrogen) containing 20% FCS, 2 mM l-glutamine, and 100 µg/ml penicillin and streptomycin). Imaging was performed at 37°C in a microscope-body-enclosing incubator. Cells on carbon-coated sapphire disks were observed by confocal microscopy (LSM510Meta or LSM780; Carl Zeiss, Oberkochen, Germany) using 10 × 0.3 NA Plan-Neofluar or 20 × 0.8 NA Plan-Apochromat objective (Carl Zeiss). The cell division process was monitored every 24 s by time-lapse imaging. For three-dimensional (3D) time-lapse imaging, cells were observed by confocal microscopy (LSM780) using 63 × 1.4 NA Plan-Apochromat objective (Carl Zeiss). For *Figure 1—figure supplement 1*, fluorescent chromatin and Lap-2α were recorded under the following conditions: 25 optical sections, section thickness of 2.0 µm, z-stacks of every 1.0 µm, the xy resolution of 0.13 µm, and a time-lapse interval of 30 s. For *Figure 3—figure supplement 2*, fluorescent chromatin was monitored under the following conditions: 40 optical sections, section thickness of 1.4 µm, z-stacks of every 0.7 µm, the xy resolution of 0.13 µm, and a time-lapse interval of 10 min. For *Figure 4*, DNA was stained with 0.2 µM silicon–rhodamine Hoechst (*Lukinavicius et al., 2015*), and the nucleus and nucleoporins were monitored under the following conditions: 25 optical sections, section thickness of 2.5 µm, z-stacks of every 1.25 µm, the xy resolution of 0.25 µm, and a time-lapse interval of 1 min. Fluorescence images were filtered with a median filter (kernel size: 0.25 × 0.25 µm) for presentation purposes.

## Segmentation of the nucleus and core regions

A 3D computational pipeline was developed in MATLAB (The MathWorks Inc., Natick, MA) that segments chromosomes and core regions from H2B-mCherry and Lap-2α-YFP channels, respectively and extracts different parameters. Original stacks were interpolated along z axis to obtain isotropic resolution and facilitate true 3D image analysis. A 3D Gaussian filter was applied to reduce the effects of high frequency noise. To detect chromosome regions, H2B-mCherry and SiR-Hoechst channels were binarized first using a multi-level thresholding method as described in *Heriche et al. (2014)*. Then, chromosome region of interest typically in metaphase in the first time point of the sequence was detected by analyzing the volume and location information of the connected components in the binary image. The detected chromosome was then tracked over the subsequent time points of the sequence and both of the daughter chromosomes were tracked after the division. The surface area of the chromosome was computed applying the method described in *Legland et al. (2007)*. For Lap-2α-YFP channel, a reference threshold was estimated by analyzing the intensity over time. This reference threshold was then adapted with a second threshold obtained from individual time points in order to segment the protein. The portion of the nuclear surface where Lap-2α localizes was marked to estimate the surface area of the core regions. Inner- and outer-core regions within nuclei were determined by dividing each nucleus with a cutting plane. The cutting plane was constructed from two vectors where the first one was directed towards the maximum elongation of nucleus and the second one was orthogonal to the first vector and was directed towards the upward z direction. These axes were determined by Eigen vector analysis on the pixel coordinates of the detected nucleus. For the measurement of the Nup intensity on the NE, segmented nuclear volume was dilated and eroded in 3D to define a nuclear membrane rim with 0.75 μm width. The areas of inner- and non-core regions were adjusted in individual time points based on the total surface area of the nuclei. Visualization of the chromosome surface in 3D was done in the Amira software package (*Pruggnaller et al., 2008*).

## Sample preparation for electron microscopy

Cells at different cell-cycle stages were instantly frozen using a high-pressure freezing machine (HPM 010; ABRA Fluid AG, Widnau, Switzerland). Just before freezing, cells were immersed in IM containing 20% Ficoll (PM400; Sigma Aldrich) for protecting cells from freezing damage. Freeze substitution into Lowicryl HM20 resin (Polysciences Inc., Warrington, PA) was performed as described in a previous report (*Kukulski et al., 2011*), with the following modifications: Frozen cells were incubated with 0.1% uranyl acetate (UA) in acetone at -90°C for 20–24 hr and, after infiltration into Lowicryl resin and UV-polymerization, samples were further polymerized by sunlight for 3–4 days. The cells were also embedded in EPON resin (Serva, Heidelberg, Germany) for enhancing membrane contrast as follows: Frozen cells were incubated with 1.0% osmium tetroxide ($OsO_4$), 0.1% UA, and 5% water in acetone at $-90$°C for 20–24 hr. The temperature was raised to $-30$°C (5°C/hour), kept at $-30$°C for 3 hr, and raised to 0°C (5°C/hr). Samples were then washed with acetone, infiltrated with increasing concentrations of EPON in acetone (25, 50 and 75%), embedded in 100% EPON and polymerized at 60°C for 2 days. Sections of 300 nm and 50 nm thickness were cut with a microtome (Ultracut UCT; Leica, Wetzlar, Germany) and collected on copper–palladium slot grids (Science Services, München, Germany) coated with 1% Formvar (Plano, Wetzlar, Germany).

## Electron tomography

As fiducial markers, 15 nm of gold-conjugated Protein A (CMC university Medical Center Utrecht, Utrecht, Netherlands) was absorbed on both sides of 300 nm sections. Sections were post-stained with 2% UA and lead citrate. Single or dual axis tilt series were acquired with a TECNAI TF30 transmission EM (TEM; 300 kV; FEI, Hillsboro, OR) equipped with a 2k x 2k Eagle camera (FEI) by using the Serial EM software (*Mastronarde, 2005*). The samples were pre-irradiated by an electron beam to minimize sample shrinkage during tilt series acquisition. Images were recorded over a $-60$° to 60° tilt range with an angular increment 1° at a pixel size of typically 0.75 nm or 1.0 nm. Tomograms were reconstructed using R-weighted backprojection method implemented in the IMOD software package (version 4.5.6) (*Kremer et al., 1996*). Dual axis tilt series were aligned using gold fiducial markers while single axis tilt series were aligned by patch tracking. It should be noted that tomographic resolution permits great advantages over classical approaches in which the thickness of a

section is in the range of the diameter of a nuclear pore and it is thus not clear which parts of it are within the section and into which exact direction they are projected into in the electron micrograph. In contrast, 3D data permit a much more accurate mapping of orientation, membrane topology and substructures for each individual NPC, which is essential to study rare intermediates.

### Immuno-electron microscopy

Grids carrying 50 nm sections were pretreated with 0.1% Trition X-100 (Sigma Aldrich) in phosphate-buffered saline (PBS) for 10 min and blocked with 1% BSA and 0.1% fish skin gelatin (Sigma Aldrich) in PBS for 1 hr. Sections were then incubated with a primary antibody mAb414 (Covance, Princeton, NJ; RRID:AB_10063490), which recognizes four nucleoporins (Nups 358, 214, 153, and 62), for 2 hr, a rabbit anti-mouse secondary antibody (Cat. No. Z0259; Dako, Hamburg, Germany; RRID:AB_2532147) for 1 hr, and 10 nm of gold-conjugated Protein A (CMC university Medical Center Utrecht) for 30 min. The antibodies and Protein A beads were diluted in PBS with 0.2% BSA and the sections were washed for five times with PBS containing 0.2% BSA between steps. After multiple washes with PBS, sections were fixed in 2.5% glutaraldehyde in PBS for 20 min in order to immobilize the antibodies and Protein A-gold beads on sections. After washing with water, sections were post-stained with 2% UA and lead citrate for contrast enhancement. All steps were carried out at room temperature. Images were taken on a TEM (CM 120 Biotwin; Phillips, Hillsboro, OR). For specificity analysis of immuno-EM labeling, the number of gold particles on assembly intermediates and ones nonspecifically attached within 50 nm under the inner nuclear membrane was counted.

### Cryo-electron tomography of isolated nuclear envelope

The nuclear envelope of HeLa cells was isolated and cryo-fixed as described previously (*Bui et al., 2013*; *Ori et al., 2013*). Tilt series of cryo-EM images were acquired using a Titan Krios TEM (FEI) at a pixel size of 0.34 or 0.43 nm and tomograms were reconstructed using the IMOD software package as described in (*Bui et al., 2013*).

### Membrane profile analysis and measurement of nuclear pore diameter

The outlines of outer and inner nuclear membrane (ONM and INM) were manually marked by clicking points within the tomographic volume in the IMOD software package. The sets of clicked points were aligned to share an x-axis corresponding to INM and interpolated using a spline fit, and the resulting coordinates were fitted locally using a second degree polynomial fit as described in *Kukulski et al. (2012)*. The maximum depth of the INM evagination was determined from these two-dimensional profiles. For the ONM/INM distance, the median of the distance at 50 points between 45 and 90 nm away from mature pores was measured. The alignment, the interpolation, and the extraction of the parameter were done in MATLAB 7.4. The maximum diameter of assembly intermediates and mature pores was measured manually from the top view images as illustrated in *Figure 2D*. The mature pores used for the measurement were the ones found in cells at >3 hr post anaphase. Unpaired t-tests with the assumption of equal variances were performed to compare two groups.

### Measurement of nuclear pore density

The number of mature pores and intermediates was counted manually in the tomograms. The nuclear surface area was measured in each tomogram by manually tracing the NE using the IMOD software package. The shrinkage of the specimen was corrected by comparing the diameter of mature pores in EM tomograms of plastic resin with the one in cryo-EM tomograms. The shrinkage was 22 ± 2.7% (the average and standard deviation, $N$ = 13 sections) and 15 ± 2.8% ($N$ = 11 sections) in Lowicryl and EPON resin, respectively. Since non-core and core regions are hard to distinguish in the matured NE after late G1, the pore density in cells >3 hr post anaphase was measured in any regions of the NE. Kinetic modeling of nuclear pore density is described in Materials and methods.

### Kinetic modeling of nuclear pore densities

We modeled pore maturation using delay equations (*Figure 3—figure supplement 1A*). Assembly intermediates are generated with a rate $V(t)$ and enter maturation with a rate constant $k_M$. After a

time interval $K_M$ an intermediate becomes fully matured to a NPC. We denote by $\tau_S$ the time required post anaphase to seal the NE and end the postmitotic assembly of the NPC. The simulation time $t$ is related to the time after anaphase onset $t_{AO}$ by $t = t_{AO} - \tau_S$. For the data shown in *Figure 3C,D* we took $\tau_S$ = 10 min by assuming a start of interphase assembly to be 10 min after anaphase onset when the NE is sealed (*Dultz et al., 2008*; *Otsuka et al., 2014*). We simulated the process for $\tau_S$ = 0–15 min, and found that $\tau_S + \tau_M$ differed by less than 1%. The number of intermediates $I$ and mature pores $M$ is given by

$$\frac{dI(t)}{dt} = V(t) - k_M I(t) \tag{1}$$

$$\frac{dM(t)}{dt} = k_M I(t - \tau_M) - k_d M(t), \tag{2}$$

where $k_d$ is the degradation rate constant of mature pores. Consequently the number of intermediates in the process of maturation $I_m$ is given by

$$\frac{dI_m(t)}{dt} = k_M I(t) - k_M I(t - \tau_M). \tag{3}$$

The total number of intermediates $I_T$ is the quantity we can measure which is given by

$$I_T(t) = I(t) + I_m(t). \tag{4}$$

Surface densities are computed from $i_T = I_T/A$, $m = M/A$, where $A$ is the nuclear surface area. The overall maturation time is defined as

$$T_M = \frac{1}{k_M} + \tau_M \tag{5}$$

We assume isotropic expansion of the nucleus where the surface area can be described by

$$A(t_{AO}) = a_0 + a_1 \left(1 - \exp\left(-k_{g1} t_{AO}\right)\right) + k_{g2} t_{AO} \tag{6}$$

We obtained $a_0$= 424 μm$^2$ (95% confidence interval (CI) [212–448]), $a_1$ = 161 μm$^2$ (95% CI [81–185]), $k_{g1}$ = 0.0722/min (95% CI [0.036–0.093]), $k_{g2}$ = 0.397/min (95% CI [0.199–0.405]) by fitting *Equation 6* to the data in *Figure 3—figure supplement 2*. We tested different intermediate production variants (*Figure 3—figure supplement 1B*) as

$$V(t) = \begin{cases} vA(t), \text{Variant 1} \\ (v_1 exp(-k_v(t - t_S)) + v_0)A(t), \text{Variant 2} \\ i(t_S)A(t_S)\delta(t - t_s) + v_0 A(t), \text{Variant 3.} \end{cases} \tag{7}$$

Variant 1 assumes a constant production rate density; Variant 2 assumes a time dependent production that decreases with time to a basal rate $v_0$; finally in Variant 3 the majority of pores are initiated at $t_S$. For Variant 3 the initial densities $i(t_S)$ and $m(t_S)$, are estimated for the inner- and outer-core regions separately. The value of $i(t_S)$ is set to 0 for Variant 2. We take $i_m(t_S)$ = 0. The system of equations (*Equations 1–6*) is solved analytically to obtain the densities of intermediates and mature pores. The production and degradation rates are estimated from this and previous studies. For the mature pore degradation rate constant we take $k_d$ = 0.00042/min, which yields a pore life time of ~40 hr (*Rabut et al., 2004*; *Schwanhausser et al., 2011*). For Variant 2 and Variant 3 we take $v_0$ = 0.0015 intermediates/μm$^2$/min. This yields an NPC density in mature NEs (3–20 hr post anaphase) of 11.47 ± 1.33 NPCs/μm$^2$ (0.65 ± 6e-4 intermediates/μm$^2$) for Variant 3 and 12.1 ± 1.2 NPCs/μm$^2$ (0.41 ± 1.36e-04 intermediates/μm$^2$) for Variant 2. Here the mean and standard deviation were estimated from inner-core*0.68 + outer-core*0.32 since the ratio of the surface area between inner- and outer-core regions is 0.68:0.32. The other model parameters are estimated by minimizing the sum of squared residuals (MATLAB routine *lsqnonlin*)

$$F = \frac{1}{\sigma^2} \sum_{j=1}^{n/2} \left(i_T\left(t_j\right) - D_i\left(t_j\right)\right)^2 + \left(m\left(t_j\right) - D_m\left(t_j\right)\right)^2, \tag{8}$$

where $D_i$ and $D_m$ are the measured densities of intermediate and mature pores (*Figure 3C,D*), $n$ is the number of data points, and $\sigma^2 = 2.18$ pores/µm² is the mean standard deviation estimated from all measurements. For Variant 1 we obtained $k_M = 7.13$/min (95% CI [0.0379–7.5730]), $\tau_M = 18.23$ min (95% CI [0.725*–27.59]) and $T_M = 18.37$ min (95% CI [16.12–27.69]), for Variant 2 $k_M = 0.0996$/min (95% CI [0.0274–2.06]), $\tau_M = 17.75$ min (95% CI [0.725*–44.59]) and $T_M = 27.78$ min (95% CI [18.26–44.72]), for Variant 3 $k_M = 1.357$/min (95% CI [0.0408–20*]), $\tau_M = 43.03$ min (95% CI [18.78–49.86]) and $T_M = 43.76$ (95% CI [41.32–50]). For Variant 2 and 3 the model fit does not change for very low values $\tau_M$ or high values of $k_M$, respectively. The asterisk indicates that the 95% boundary of the distribution has not yet been reached at the given value. The profile-likelihood method (*Venzon and Moolgavkar, 1988*) has been used to estimate the 95% confidence. In this method the log-increase $\varphi(par) = n\left[\log\left(\frac{F(par)}{n}\right) - \log\left(\frac{F(par_{min})}{n}\right)\right]$ of the mean squared distance $F$ with respect to the best fit $par_{min}$ was computed by varying the parameter of interest and optimizing the other parameters to the $n$ data points. For $|\varphi(par)| < \chi^2_{1,0.95} = 3.84$ the parameter is within its 95% CI. For $T_M$ (*Equation 5*) the confidence interval is computed from the values of $k_M$ and $\tau_M$. The quality of the fits (lower sum of squared residuals, *Figure 3—figure supplement 1C*) was slightly better for Variant 3. Furthermore the obtained maturation time was more in agreement with previous reported values (*Dultz and Ellenberg, 2010*). We thus investigate alternatives to the maturation mechanism using Variant 3 only.

The model with a multi-step maturation process (*Figure 3—figure supplement 1F*) reads

$$\frac{dI_1}{dt} = V(t) - k_{M1}I_1 \tag{9}$$

$$\frac{dI_j}{dt} = k_{M(j-1)}I_{j-1} - k_{Mj}I_j, \quad \text{for } j = 2, ..., N-1 \tag{10}$$

$$\frac{dM}{dt} = k_{M(N-1)}I_{N-1} - k_d M. \tag{11}$$

The sum of all intermediates $\sum_{j=1}^{N-1} I_j/A$ and $M/A$ are fitted to the intermediate and mature pore densities, respectively. We modeled Variant 3 for the pore initiation. We found that there was no significant difference in the quality of the fits when assuming equal transition rate constants. Consequently the simulations shown are for $k_{Mj} = k_M$. The maturation time defined as the characteristic time of mature pore appearance reads

$$T_M = \sum_{j=1}^{N-1} \frac{1}{k_{Mj}} = \frac{(N-1)}{k_M}. \tag{12}$$

We simulate different maturation times by allowing a maturation time distribution $P(\tau)$, with finite positive support as

$$\frac{dI(t)}{dt} = V(t) - k_M I(t) \tag{13}$$

$$\frac{dI_m(t)}{dt} = k_M I(t) - k_M \int_0^\infty P(\tau)I(t-\tau)d\tau \tag{14}$$

$$\frac{dM(t)}{dt} = k_M \int_0^\infty P(\tau)I(t-\tau)d\tau - k_d M(t). \tag{15}$$

The example shown in *Figure 3—figure supplement 1H* is for an uniform distribution of $\tau_M \pm w$, where $w$ is the half-width of the distribution.

## Genome editing

For tagging Nup107 at the N-terminus with monomeric enhanced GFP (mEGFP), zinc finger nucleases (ZFN) containing DNA binding sequences in the 5′-3′ direction of TCAGTACTGATG and GCTGAGCCCGAAGTC were purchased from Sigma Aldrich. The donor plasmid consists of mEGFP cDNA sequence flanked by a left homology arm (ENSEMBL release 75, ENST00000229179,

chromosome 12: 68686269–68687065) and a right homology arm (ENSEMBL release 75, ENST00000229179, chromosome 12: 68687065–68687892). ZFN and the donor plasmid were transfected into HeLa cells as described in *Mahen et al. (2014)*. For tagging Nup358 at the N-terminus with mEGFP, CRISPR-Cas9 nickases were used. pX335-U6-Chimeric_BB-CBh-hSpCas9n(D10A) was a gift from Feng Zhang (Addgene plasmid # 42335, Cambridge, MA), and gRNAs were designed using the Feng Zhang Lab's Target Finder (http://crispr.mit.edu/). The following gRNAs for Nup358 were cloned into pX335 (*Cong et al., 2013*): 5'CCTGAGCGCTGGTCTCACGCGCC3' and 5'GAGGCGCAGCAAGGCTGACGTGG3'. CRISPR-Cas9 nickases and the donor plasmid were transfected using jetPRIME reagent (Polyplus, New York, NY), according to the manufacture's protocol. 7–10 days after transfection, cells were sorted with a MoFlo Legacy cell sorter (Beckman Coulter, Brea, CA) as described in *Mahen et al. (2014)*.

## Junction PCR

Genomic DNA was prepared using ISOLATE II Genomic DNA Kit (Bioline, Taunton, MA) according to the supplier's manual. Junction PCR was performed at endogenous loci to detect the insertion of mEGFP using separate sets of primers, one of which anneals inside mEGFP and the other one outside of the gene of interest. The primer sequences are as follows: Nup107 forward (5'ATTAATAAAAGGTATAAATGCCAGCAACAG3'), Nup107 reverse (5'CACCTGGTCAACAACTACTTACTCCT3'), NUP358 forward (5'GCATAAGACGGTGGTTCTGGAACCAATC3'), and NUP358 reverse (5'AGCAAACTGACTCAAGATTCTGCGCA3'). Touchdown PCR was performed using HotStar HiFidelity (Qiagen, Hilden, Germany) according to the supplier's protocol.

## Western blot

Cells were lysed for 20 min on ice in lysis buffer (10% glycerol, 1 mM DTT, 0.15 mM EDTA, 0.5% Triton X-100, complete protease inhibitor cocktail and PhosSTOP (Roche, Basel, Switzerland)). Protein concentration was quantitated using the Bio-Rad Protein Assay (Bio-Rad, Hercules, CA). 40 µg of total protein was run onto NuPAGE®4–12% Bis-Tris Gels (Novex Life Technologies, Waltham, MA) and transferred onto PVDF membrane using the Bio-Rad transfer system. After blocking with 5% milk solution (nonfat milk powder in PBS + 0.1% Tween 20), the following primary antibodies were used to label the proteins of interests: anti-Nup107 (ab178399, abcam, Cambridge, United Kingdom; RRID:AB_2620147), anti-RanBP2 (ab197044, abcam; RRID:AB_2620148), anti-tubulin (DM-1A, Sigma; RRID:AB_521686) and anti-GFP (Cat. No. 11814460001, Roche; RRID:AB_390913). Subsequently horseradish peroxidase (HRP)-conjugated secondary antibodies (ECL anti-rabbit IgG HRP-linked whole antibody NA934V; RRID:AB_772206, or ECL anti-mouse IgG HRP-linked whole antibody NA931V; RRID:AB_772210, GE Healthcare, Little Chalfont, United Kingdom) were used to detect the protein of interests with chemiluminescence reaction.

## Kinetic analyses of Nup107 and Nup358 assembly

Average intensities of Nup358 and Nup107 in the inferred inner-core and non-core regions were quantified. The total intensities were calculated by multiplying the average intensities by the nuclear surface area. Methods for the segmentation of core regions is described in 'Segmentation of the nucleus and core regions' section above. We formulated a sequential assembly model that describes the recruitment of Nup107 and Nup358. Nup107 accumulates first with a rate constant *k*, and Nup358 assembles later with a rate constant *l*. The number of NPC intermediates can be described as

$$\frac{dN_0^x}{dt} = -k_x N_0^x \tag{16}$$

$$\frac{dN_1^x}{dt} = k_x N_0^x - l_x N_1^x, \quad \text{where } x = pm, ip \tag{17}$$

$$\frac{dN_2^x}{dt} = l_x N_1^x, \tag{18}$$

where $N_{0,1,2}^{pm,ip}$ denote the number of NPCs that assemble through the interphase (*ip*) or postmitotic (*pm*) pathway without Nup107 nor Nup358 ($N_0^{pm}$ and $N_0^{ip}$), with Nup107 only ($N_1^{pm}$ and $N_1^{ip}$), or both Nup107 and Nup358 ($N_2^{pm}$ and $N_2^{ip}$). The rate constants for postmitotic and interphase recruitment of

Nup107 and Nup358 are given by $k_{pm}$ and $k_{ip}$, and $l_{pm}$ and $l_{ip}$, respectively. The degradation and interphase production rates are small (*Rabut et al., 2004*; *Dultz and Ellenberg, 2010*; *Schwanhausser et al., 2011*) and can be neglected in the time frame of 2 hr post anaphase (see also 'kinetic modeling of nuclear pore densities' above). The total number of NPCs containing Nup107 and Nup358 are then given by

$$T_{Nup107} = \left[ \left( N_1^{pm} + N_2^{pm} \right) f_{pm} + \left( N_1^{ip} + N_2^{ip} \right) \left( 1 - f_{pm} \right) \right] \tag{19}$$

$$T_{Nup358} = \left[ N_2^{pm} f_{pm} + N_2^{ip} \left( 1 - f_{pm} \right) \right] \tag{20}$$

where $f_{pm}$ is the fraction of postmitotic NPC. In the non-core region $f_{pm} = 0.92$, whereas in the inner-core region $f_{pm} = 0.5$ (*Figure 3*). Since the nuclear membrane is not yet sealed before 10 min post anaphase (*Dultz et al., 2008*; *Otsuka et al., 2014*), we take for initial condition $N_0^{ip}(t = 10 \text{ min}) = 1$, and 0 for $t < 10$ min. For the postmitotic assembly we take $N_0^{pm}(t = 4 \text{ min}) = 1$, and 0 for $t < 4$ min. The four kinetic rate constants $k_{pm}$, $k_{ip}$, $l_{pm}$ and $l_{ip}$ are estimated by simultaneously fitting the total Nup107 and Nup358 intensities in non-core and inner-core regions from 4 min up to 125 min post anaphase (*Figure 4*). To match the experimental data normalization we also normalize the simulations. The normalization coefficients range from 1–1.1. The normalized pore densities are obtained by multiplying the normalized total number of pores by the nuclear surface area and subsequently dividing it by the maximal area. We obtained $k_{pm} = 0.355$/min (95% CI [0.333–0378]), $l_{pm} = 0.0437$/min (95% CI [0.0425–0.0449]), $k_{ip} = 0.0374$/min (95% CI [0.0335–0.0417]), and $l_{ip} = 0.0276$/min (95% CI [0.0209–0.0354]). Confidence intervals are obtained as explained in 'kinetic modeling of nuclear pore densities' above.

## Stimulated emission depletion (STED) microscopy

After release from thymidine block, the division process of GFP-Nup107 genome-edited cells were monitored every 30 s by confocal microscopy (LSM780; Carl Zeiss) using 10 × 0.3 NA Plan-Neofluar objective (Carl Zeiss). Cells were then fixed with paraformaldehyde and immunostained as described in the previous report (*Szymborska et al., 2013*), with rabbit anti-Nup358 (Cat. No. HPA018437, The Human Protein Atlas; RRID:AB_2620151) and mouse anti-GFP (Cat. No. 11814460001, Roche; RRID:AB_390913) antibodies, and Abberior STAR RED-conjugated anti-rabbit IgG (Cat. No. 2-0012-011-9, Abberior GmbH, Göttingen, Germany; RRID:AB_2620152) and Abberior STAR 580-conjugated anti-mouse IgG (Cat. No. 2-0002-005-1, Abberior GmbH; RRID:AB_2620153). Cells were mounted in Vectashield containing 4′,6-diamidino-2-phenylindole (DAPI) (Cat. No. H-1500, Vector Laboratories Inc., Burlingame, CA). Super-resolution imaging was performed on a Leica SP8 3X STED microscope, equipped with 775 nm pulsed wave depletion and white light pulsed lasers, Leica HCX 100 × 1.4 NA Plan Apochromat objective, and time-gated hybrid detectors (Leica HyD). Excitation wavelength was adjusted to 580 and 633 nm, and bandpass filters were set to 585−630 and 650−702 nm, and the two channels were recorded pseudo-simultaneously by line switching. The fluorescent nuclei stained with DAPI were also recorded afterwards. The images were taken with a final optical pixel size of 20 nm, z-stacks of every 200 nm, and the optical section thickness of 550 nm. Images were filtered with a Gaussian filter (kernel size: 0.5 × 0.5 pixel) for presentation purposes.

## Quantification of STED data

Lines with the width of 400 nm were drawn along the edge of the DAPI-stained nuclei. Non-core and core regions were inferred as described in *Figure 1—figure supplement 1B–G*. The NEs on the lines were flattened and fluorescence intensity was quantified after binning of 15 pixels (correspond to 300 nm width) along the lines. The intensity difference between two channels was normalized using the images in non-core regions, and the intensity ratio of Nup107 to Nup358 was measured. All analyses were done in ImageJ (http://rsbweb.nih.gov/ij/).

## Particle averaging of mature pores and pore intermediates

Assembly intermediates which have similar membrane profiles were selected at each time point and subjected to subtomogram averaging. The averaging was done on nuclear pores in freeze-substituted and plastic-embedded cells using the previously described averaging method (*Beck et al.,*

*2004*). Briefly, the subtomograms, which contain mature pores and intermediates, were extracted from the tomograms. The extracted subtomograms were aligned using iterative missing wedge compensation alignment procedure. Afterwards, the aligned subtomograms were averaged and visualized. The mature pores used for the averaging are the ones found in cells at >3 hr post anaphase. The overall structural similarity of the averaged nuclear pores to the respective cryo structures (*Figure 1E*) indicates a good structure preservation in freeze-substituted and plastic-embedded cells.

## Sample size determination and statistical analysis

For correlative light and electron microscopy, we first obtained one tomogram in each non-, inner- and outer-core region in 4 different cells at 19, 28, 53, and 116 min after anaphase onset as pilot experiments. We then increased the number of dataset and eventually took 158 tomograms in 14 different cells. The exact value of the analyzed surface area and the number of nuclear pores found are listed in *Table 1*. We picked up all the NE evaginations which were visible in the EM tomograms and did not perform any selection. Statistical analyses of the pore structure and density were performed only after all the data were taken. For immuno-EM, time-lapse 3D imaging, and STED microscopy, the data were from two independent experiments and the statistical analysis was carried out after all the data were obtained. Statistical analysis methods, sample sizes (N) and P values (P) for each experiment are indicated in figure legends.

## Acknowledgements

We thank the European Molecular Biology Laboratory Electron Microscopy Core Facility (especially R Mellwig and C Funaya) and Advanced Light Microscopy Facility (especially M Lampe and S Terjung); B Klaus for help with the statistical analysis; M Kueblbeck for help with generating genome-edited cells; S Mosalaganti for help with subtomogram averaging; the members of the Ellenberg group and the Beck group for advice and discussion. The work was supported by: the European Molecular Biology Laboratory and the EMBL Interdisciplinary Postdoc Programme under Marie Curie Actions COFUND (SO), funding from the German Research Council to JE (DFG EL 246/3-2 within the priority program SPP1175), and funding from the European Research Council to MB (309271-NPCAtlas). SO was additionally supported by a JSPS fellowship (postdoctoral fellowship for research abroad). KHB was supported by postdoctoral fellowships from the Swiss National Science Foundation, the European Molecular Biology Organization and Marie Curie Actions.

## Additional information

### Funding

| Funder | Grant reference number | Author |
| --- | --- | --- |
| European Commission | Marie Curie Actions COFUND | Shotaro Otsuka |
| Japan Society for the Promotion of Science | Postdoctoral Fellowship for Research Abroad | Shotaro Otsuka |
| Schweizerischer Nationalfonds zur Förderung der Wissenschaftlichen Forschung | | Khanh Huy Bui |
| European Molecular Biology Organization | | Khanh Huy Bui |
| European Commission | Marie Curie Actions | Khanh Huy Bui |
| European Research Council | 309271-NPCAtlas | Martin Beck |
| Deutsche Forschungsgemeinschaft | SPP1175 EL 246/3-2 | Jan Ellenberg |

The funders had no role in study design, data collection and interpretation, or the decision to submit the work for publication.

## Author contributions
SO, Designed the project, Performed all the experiments and analyses except for cryo-EM tomography, segmentation of fluorescence images, kinetic modeling, and generation of genome-edited cell lines, Wrote the paper; KHB, Performed cryo-EM tomography and assisted with subtomogram averaging, Analysis and interpretation of data; MS, Contributed to the computational quantitative analysis of EM images; MJH, Carried out the segmentation of fluorescence images, Analysis and interpretation of data; AZP, Contributed to kinetic modeling, Analysis and interpretation of data; BK, Generated genome-edited cell lines; ME, Helped with the optimization of EM experiments, Analysis and interpretation of data; MB, Designed the project, Supervised the work, Analysis and interpretation of data, Drafting or revising the article; JE, Designed the project, Supervised the work, Wrote the paper, Analysis and interpretation of data

## Author ORCIDs
Shotaro Otsuka, http://orcid.org/0000-0003-3976-0843
M Julius Hossain, http://orcid.org/0000-0003-3303-5755
Jan Ellenberg, http://orcid.org/0000-0001-5909-701X

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
