## [Decision Letter]

Thank you for submitting your article "Nuclear pore assembly proceeds by an inside-out extrusion of the nuclear envelope" for consideration by *eLife*. Your article has been assessed by Karsten Weis as Reviewing Editor and Ivan Dikic as the Senior Editor.

The editors have discussed their own reviews and three reviews that were available from another journal with one another and the Reviewing Editor has drafted this decision to help you prepare a revised submission.

In this manuscript, Beck, Ellenberg and colleagues combine high-resolution electron microscopy, supper-resolution microscopy and correlative light microscopy to characterize how nuclear pore complexes assemble and insert into the nuclear envelope. Their analysis reveals various dome shaped intermediates at the inner nuclear membrane suggesting the NPC assembly occurs by an "inside-out extrusion" mechanism.

With the manuscript, the authors also provided three reviewers' comments from a prior submission to another journal. In summary, the three reviewers praise the high technical quality of the data but there is a concern that the data are largely descriptive and do not provide molecular insight into the mechanism of how NPCs insert into the double membrane of the nuclear envelope. However, while the reported observations are indeed largely descriptive in nature, the data described in this manuscript provide an important framework for the understanding of interphase NPC assembly and therefore should be of interest to a large audience. This manuscript clearly goes beyond previous studies in yeast or *Xenopus* and hence should be published in *eLife*. As pointed out by the previous reviewers, the data are in general of high technical quality but some of the technical issues, that were brought up in the prior reviews, need to be addressed in a revision. This includes a test of the robustness of the model.

---

## [Author Response]

In the present study, we could for the first time capture the structural intermediates of the assembly process by which nuclear pore complexes are inserted into the double membrane barrier of intact nuclei in human cells. We achieved this by correlating live imaging of cells at different stages of nuclear growth with high-resolution 3D electron and super-resolution fluorescence microscopy. Systematic EM tomography and its quantitative structural analysis allowed us to reconstruct the mechanism of nuclear pore biogenesis as an inside-out extrusion of the nuclear envelope. Super-resolution fluorescence microscopy furthermore allowed us to determine the molecular maturation of the assembly intermediates. Together, our data show that nuclear pores assemble symmetrically, by an evagination of the inner nuclear membrane that grows in diameter and depth until it touches and fuses with the flat out r nuclear membrane. From the beginning, the evagination is supported by the eight-fold symmetric nuclear ring complex and driven by a novel mushroom-shaped protein complex.

The main concern, especially of reviewer #1, was that the assembly mechanism we demonstrated was not sufficiently novel. We respectfully disagree with this opinion. While it is true that classic studies in yeast mutants that have proposed many different models for NPC assembly have also mentioned an inside-out process as a possibility in the past, these studies never demonstrated which model actually takes place or what the assembly intermediates might look like. Similarly, the work on in vitro assembled nuclei in *Xenopus* egg extracts, which revealed a requirement for nuclear import of Nup153 for NPC assembly did not demonstrate by what process or through which structural intermediates NPC assembly takes place.

Our study remains the first that actually resolves the progressive membrane deformation of NPC assembly intermediates and demonstrates that it is the inner nuclear membrane that is evaginated towards the outer nuclear membrane, which on the contrary remains flat until fusion. It furthermore for the first time shows that the eight-fold symmetric nuclear ring complex is preassembled to support the membrane deformation by a novel mushroom shaped protein density. This “inside-out” mechanism, demonstrated in situ in human cells, provides a completely new mechanistic framework to interpret existing genetic (yeast) and biochemical (*Xenopus*) data and is therefore in our opinion a conceptually very important advance for the field and for cell biology in general.

We have addressed the comments of the previous reviewers and provided technical clarifications in our revisions. In particular, we have tested the robustness of our model for pore assembly, as requested by two of the previous reviewers.

We have modeled the different scenarios for the appearance of precursors and mature pores.

1) A continuous production of pore precursors (Variant 1)

2) A time dependent production (Variant 2)

3) A sudden burst of pore precursors (Variant 3)

We found that models where the majority of intermediates are produced within 25 min after anaphase onset (Variant 2, Variant 3) fit the data well whereas a model with a constant production only (Variant 1) did not fit (Figure 3—figure supplement 1). Although the fits to the data did not allow distinguishing between a sharp burst of intermediates or a more gradual decay of a postmitotic increase in their production, either model clearly showed that the number of observed assembly intermediates can quantitatively explain the increased number of mature pores observed later (Figure 3—figure supplement 1). Since the model with Variant 3 fit the data best, we used Variant 3 for our modeling.

To account for potential heterogeneity in the maturation process, we have simulated multiple intermediate steps as well as a broader distribution of maturation times. Similar average maturation times were obtained even if several maturation steps were explicitly included (Figure 3—figure supplement 1) or the maturation time was allowed to have a broader distribution (Figure 3—figure supplement 1), demonstrating the robustness of our results. For a large number of maturation steps the multi-step model converged to the fixed delay model (Figure 3—figure supplement 1). This indicates that the maturation process likely requires several intermediate steps and that the delay model represents a simplified description for such a process.

These tests confirmed that our results based on the modeling are robust and that we are using the model that best fits the experimental data. You can find these revisions in a new figure, Figure 3—figure supplement 1, as well as in the Results (subsection “Abundance of intermediates matches increase in mature pores during nuclear growth”) and the Materials and methods (“Kinetic modeling of nuclear pore densities”).